# Development and Validation of a Deep-Learning-Based Algorithm for Detecting and Classifying Metallic Implants in Abdominal and Spinal CT Topograms

**DOI:** 10.3390/diagnostics14070668

**Published:** 2024-03-22

**Authors:** Moon-Hyung Choi, Joon-Yong Jung, Zhigang Peng, Stefan Grosskopf, Michael Suehling, Christian Hofmann, Seongyong Pak

**Affiliations:** 1Department of Radiology, Eunpyeong St. Mary’s Hospital, College of Medicine, The Catholic University of Korea, Seoul 03312, Republic of Korea; cmh@catholic.ac.kr; 2Department of Radiology, Seoul St. Mary’s Hospital, College of Medicine, The Catholic University of Korea, Seoul 06591, Republic of Korea; 3Siemens Medical Solutions USA, Inc., Malvern, PA 19355, USA; zhigang.peng@siemens-healthineers.com; 4Siemens Healthcare GmbH, Computed Tomography, 91301 Forchheim, Germany; stefan.grosskopf@siemens-healthineers.com (S.G.); michael.suehling@siemens-healthineers.com (M.S.); hc.hofmann@siemens-healthineers.com (C.H.); 5Siemens Healthineers Ltd., Seoul 06620, Republic of Korea; seongyong.pak@siemens-healthineers.com

**Keywords:** deep learning, computed tomography, metal detection, metallic implants, topogram

## Abstract

Purpose: To develop and validate a deep-learning-based algorithm (DLA) that is designed to segment and classify metallic objects in topograms of abdominal and spinal CT. Methods: DLA training for implant segmentation and classification was based on a U-net-like architecture with 263 annotated hip implant topograms and 2127 annotated spine implant topograms. The trained DLA was validated with internal and external datasets. Two radiologists independently reviewed the external dataset consisting of 2178 abdomen anteroposterior (AP) topograms and 515 spine AP and lateral topograms, all collected in a consecutive manner. Sensitivity and specificity were calculated per pixel row and per patient. Pairwise intersection over union (IoU) was also calculated between the DLA and the two radiologists. Results: The performance parameters of the DLA were consistently >95% in internal validation per pixel row and per patient. DLA can save 27.4% of reconstruction time on average in patients with metallic implants compared to the existing iMAR. The sensitivity and specificity of the DLA during external validation were greater than 90% for the detection of spine implants on three different topograms and for the detection of hip implants on abdominal AP and spinal AP topograms. The IoU was greater than 0.9 between the DLA and the radiologists. However, the DLA training could not be performed for hip implants on spine lateral topograms. Conclusions: A prototype DLA to detect metallic implants of the spine and hip on abdominal and spinal CT topograms improves the scan workflow with good performance for both spine and hip implants.

## 1. Introduction

The number of surgical procedures using metallic implants such as total hip joint replacement and spine fusion has continuously increased [1,2]. Therefore, radiologists often encounter computed tomography (CT) images in which metallic implant artifacts considerably degrade image quality and impair diagnostic performance. In the pelvis, artifacts from hip prostheses significantly degrade the quality of the images and limit the proper evaluation of the adjacent organs [3]. Spinal implants also impair the image quality of both spinal CT images, and CT images of the chest and abdomen, for which the field of view contains the spinal regions [4,5]. Therefore, many algorithms have been developed to reduce metal artifacts and improve image quality [6].

Iterative metal artifact reduction (iMAR) is a well-established technique for reducing metal artifacts in CT examinations [7,8,9,10,11]. Because different geometric and physicochemical properties of metallic implants cause distinct metal artifact patterns, the application of iMAR algorithms should be individualized for each CT scan. Therefore, the selection of an optimal iMAR preset is a crucial step in reconstructing CT images with decent diagnostic quality. In the event of metallic implant presence, this additional step requires more time and extra attention from radiographers [9]. The omission of the iMAR application step leads to elongation of the postprocessing time due to the need for repeated reconstruction, or the lost chance to obtain iMAR-reconstructed images as a result of the limited storage time of the raw data. Moreover, improper iMAR preset selection results in a suboptimal image quality.

A topogram is an anterioposterior (AP) projection or lateral projection image that is made prior to a CT scan to determine the range of the CT scan. Aside of this purpose, topograms have typically played no other role in the examination workflow or diagnostic process. However, the use of topograms for new purposes is expanding. For example, in abdominal CT, routine inspection of topograms is helpful in detecting important findings that can be overlooked during assessment of CT images [12]. Topograms have also been used in the prediction of patient body weight or with deep learning algorithms (DLAs) in the reconstruction of three-dimensional images from two topograms for dose modulation [13,14,15]. Segmenting images of the specific organs in topograms using DLAs has also been a promising application [16,17]. However, the detection and classification of metallic implants based on topograms has not yet been studied.

The aim of this study is to develop and validate a DLA that is designed to segment and classify metallic objects in abdominal and spinal CT topograms as the first step toward the automation of iMAR-related processes.

## 2. Materials and Methods

The institutional review boards of Eunpyeong St. Mary’s hospital (hospital A) and Seoul St. Mary’s hospital (hospital B) approved this retrospective study and waived informed consent for patients due to the retrospective study design.

### 2.1. Development and Validation of the DLA

Semantic segmentation has many applications in biomedical image analysis including in the areas of CT scanning, magnetic resonance imaging (MRI) scanning, X-ray, digital pathology, microscopy and endoscope. From a technical point of view, semantic segmentation is applied here, because we obtain a 2D medical image (topogram) of N × M for which we want to generate the corresponding map of N × M × k (where k is the number of classes, including one class for the background). There are many deep learning architectures that can solve this problem [18,19,20].

Appendix A shows the architecture of our chosen neural network. First, we resample the input image to a constant matrix size (473 × 473 pixels). Based on the resampled input image, we use ResNet-101 to build a feature representation of the input image such that different classes can be separated in the feature space [21]. The 101 convolutional layers in ResNet-101 mostly have 3 × 3 filters. Second, we use 4 pooling layers with different sizes to extract the features from the global context. As the lower-dimensional features from these pool layers have different sizes, we use up-sampling with bilinear interpolation to achieve the same size as that of the original features. Finally, all the features are concatenated together as the final pooling result. Pixelwise cross entropy loss is used to evaluate the class predictions for each pixel vector individually, and gradient optimization is used to minimize prediction entropy. The results are displayed as the range of the implant, contour of the implant and number of detections per topogram (Appendix A). For training and validation of the DLA, we collected abdominal CT examinations between February 2019 and February 2021 (*n* = 30,859) and spinal CT examinations between December 2019 and December 2020 (*n* = 3650) for patients older than 16 years from a university hospital (hospital A). An abdominal radiologist with 13 years of experience reviewed the images and selected those with hip or spine implants. The final dataset consisted of 2005 CT examinations (1439 abdominal CTs with AP topograms and 566 spinal CTs with lateral topograms). These were annotated by a board-certified radiologist and considered as the ground truth. The type of metallic implants (spine or hip) and the extent of metallic implants were recorded. To avoid a potential training bias, we used a stock of CT volumes and AP topograms that had already been collected from other hospitals (*n* = 2127) to reduce potential bias, called pre-study-data for training the algorithm to detect spinal implants. A subset of CT data from hospital A (*n* = 102) as well as pre-study-data (*n* = 161) were used to train the algorithm to detect hip implants. Apart from resampling, none of the input data was transformed, but used as is for training unseen data from hospital A for validation of the DLA.

### 2.2. External Validation of the DLA on Prototype Software

External validation was performed for images of spine and hip implants. We collected consecutive CT examinations from another tertiary university hospital (hospital B) from March 2022 until we reached the target number (more than 2000 abdominal CT and more than 500 lumbar spine CT examinations) in a retrospective manner. At this institution, abdominal CT included only AP topograms, and lumbar spinal CT included both AP and lateral topograms. These were evaluated separately as three separate groups: abdominal CT_AP, spinal CT_AP, and spinal CT_Lat.

The prototype software V.9.1 (AiMAR classification Review Prototype) receives topograms as input and delivers the type (spine vs. hip) and extent of the metal based on the pixel row as output. Two radiologists (an abdominal radiologist with 13 years of experience [reader 1] and a musculoskeletal radiologist with 20 years of experience [reader 2]) independently reviewed the topograms without any annotation. They determined the cranio–caudal range of spine or hip implants, separately. When there was a discrepancy between the two radiologists, they reviewed the images together and reached a consensus.

### 2.3. Impact on Reconstruction Time by Integrating the DLA into iMAR Process

The current iMAR approach includes pre-selection of iMAR for specific types of implants, slice-wise detection of the metal, and iMAR reconstruction of the metal-containing image slice. Detection of metal and iMAR reconstruction steps requires additional time. In addition, CT images in a patient with different types of implants should be repeatedly reconstructed with different iMAR presets because only one iMAR preset can be applied throughout the scan volume. Integration of our topogram-based DLA into the iMAR process will significantly reduce reconstruction time by omitting the slice-wise metal detection step and repeated reconstruction of the entire volume with different iMAR presets in case of more than one metallic implant (Appendix A). We simulated the reconstruction time that can be achieved by integrating the DLA into iMAR processing and compared it to the current iMAR process.

### 2.4. Statistical Analysis

To validate the DLA, ground truth data were used to assess its performance. The prototype DLA provided performance metrics based on pixel rows. This 1-D metric was chosen since the relevant input for an automated iMAR reconstruction is required slice-wise along the z-axis. Thus, the input for reconstruction is read from pixel-rows of the segmented topogram and there is no impact on iMAR reconstruction from the X- or Y-direction of the segmentation mask.

The performance numbers for binary classifiers were defined for pixel rows of the topogram as described in the Appendix A. Intersection over union (IoU) is a metric that quantifies the degree of overlap between two regions of interest in two-dimensional space, as is the metric we used to measure the degree of overlap of the z-axis length of the bounding boxes determined by the DLA and the radiologists.

For external validation, the sensitivity, specificity and accuracy of the DLA were compared with those of radiologists 1 and 2 using McNemar tests. IoU was calculated between the DLA and the two radiologists in a pairwise manner. Paired t tests were used to compare the difference in IoU between the DLA and each radiologist and between the two radiologists.

A *p* < 0.05 was considered statistically significant. Statistical analyses were performed by SPSS version 23.0 (IBM, Armonk, NY, USA).

## 3. Results

### 3.1. Performance of the DLA during the Developmental Stage

Validation was performed using 1898 and 1939 CT examinations featuring spine and hip implants, respectively. The dataset included 1336 spine implants (818 shown in AP topograms, 518 in lateral topograms) and 582 hip implants shown in AP topograms. The DLA showed good diagnostic performance for both spine and hip implants (Table 1). The accuracy was higher than 98.5% for both implant types. All the other parameters were higher than 95%. The IoU of the DLA for spine and hip implants was 93.0% and 96.2%, respectively.

### 3.2. External Validation on Prototype Software

#### 3.2.1. Characteristics of the External Validation Dataset

A total of 2178 abdominal CT examinations with AP topograms and 515 lumbar spinal CT examinations with AP and lateral topograms were collected. Detailed information about the CT examinations used for external validation are summarized in Table 2. Seven different CT scanners were used to perform the examinations. Spine and hip implants were detected in 62 and 37 examinations in the abdominal CT_AP group, 244 and 17 examinations in the spinal CT_AP group, and 238 and 13 examinations in the spinal CT_Lat group, respectively. The median length of hip implants in the spinal CT AP topograms was the longest, with 412 pixel-rows.

#### 3.2.2. Abdomen CT AP Tomogram

The sensitivity of the DLA in detecting spine implants was 94.6% per pixel row and 93.6% per patient (Appendix A). The specificity of the DLA was significantly lower than that of both readers (Figure 1a). The positive predictive value (PPV) of the DLA was 85.2% (58/68), which signifies that the DLA detected 10 false positive spine implants. For hip implants, the per-pixel row sensitivity of the DLA was 78.1% despite the much higher per-patient sensitivity (36/37, 97.3%). Although the DLA showed 90% PPV for hip implants, none of the DLA parameters was significantly different from those of the readers (Figure 1b).

#### 3.2.3. Spine CT AP and Lateral Topograms

On spinal CT AP topograms, the DLA per-pixel row sensitivity was lower than the per-patient sensitivity for both spine and hip implants (94.2 versus 98.8%, 88.6% versus 94.1%) (Appendix A). The diagnostic performance of the DLA for spine implants was not significantly different from those of the radiologists, with all parameters higher than 98% (Figure 1c). For hip implants, only the specificity of the DLA (94.6%) was significantly lower than those of the readers (reader 1, 99.6%; reader 2, 100%) (Figure 1d).

On spinal CT lateral topograms, both radiologists had excellent results in detecting both spine and hip implants. The DLA showed excellent per-patient results, with all parameters higher than 99% for spine implants (Figure 1e, Appendix A). However, the DLA could not be trained to detect the hip implants at all on lateral topograms, resulting in 0% sensitivity which was significantly lower than those of the radiologists (Figure 1f).

#### 3.2.4. Retrospective Observations for DLA Misclassification

With abdominal CT topograms, the DLA frequently misclassified vertebroplasty cements (*n* = 6) as metallic spine implants (Figure 2a). Residual barium in the transverse colon (*n* = 1), chemoport (*n* = 1) and external metallic structures outside the body (*n* = 2) were other causes of false-positive classification.

With spinal CT AP and lateral topograms, the leading cause of misclassification by the DLA was sacroiliac extension of the rod-screw systems, the majority of which were interpreted as hip implants, rather than components of spine implants (*n* = 21) (Figure 2b). Generators (*n* = 2), proximal rods (*n* = 2) and penile implants (*n* = 2) follow as false-positive classification sources.

Most of the detection failure by the DLA occurred when there was no classic rod-screw instrumentation system (Figure 2c). In addition, the DLA did not detect any hip implants in lateral topograms.

#### 3.2.5. Intersection over Unit between DLA and Radiologists

The IoU values between the DLA and the radiologists as calculated from the true positive cases are summarized in Table 3. The interreader IoU values between the two radiologists were higher than 0.95 for both spine and hip implants in all three groups. In the abdominal CT_AP group, only the IoU value between the DLA and reader 1 for spine implants was significantly lower than the interreader value (*p* < 0.05). All IoU values between the DLA and each reader in spinal CT_AP and spinal CT_Lat groups were significantly lower than the interreader values. Although the values for hip implants were lowest on the spinal CT AP topogram, they were still 0.912 and 0.914.

### 3.3. Impact on Reconstruction Time by Integrating the DLA

The DLA integration into the iMAR process allows for a reduction of reconstruction times (Figure 3). Simulation calculations on 1996 topograms, in which we included the result of our DLA, show that reconstruction times can be reduced to 72.6 ± 6.8% compared to the current iMAR reconstruction. In addition, the average reconstruction time in the group of patients with two metal implants is significantly reduced to 52.4 ± 6.1%, by iMAR reconstruction in one (fused) pass.

The curves depict the relative reconstruction time based on the proportion of metal along the z-axis for volumes containing only one metallic implant (hip or spine) and both metallic implants.

## 4. Discussion

CT images near dense metallic implants are difficult to interpret due to obscuration from these artifacts. To address this problem, the iMAR algorithm that modifies projection or image data has been developed over the course of the last decade [22,23,24]. However, it should only be selectively applied to the patients because the application of iMAR is resource intensive and implant specific [25]. In this study, we developed and validated a DLA-based prototype software to detect and classify metallic implants and determine the range of implants on CT topograms. Since the ultimate goal of this study is to integrate the DLA into the CT console to automatically select the optimal iMAR presets prior to reconstruction, a prototype DLA was developed that processes topograms as input. This approach also helps to reduce reconstruction time by merging iMAR reconstructions with different presets for spinal implants and hip implants into one reconstruction. The performance of the DLA in the prototype was validated with an external test set composed of abdominal and spinal CT examinations obtained consecutively from multiple CT machines to reflect the prevalence of metallic implants. In the external validation, the DLA showed good performance for the detection and classification of spine implants on both AP and lateral topograms and hip implants on AP topograms.

Despite the different detection rates of spine implants in abdominal CT (62/2178) and in spinal CT (244/515), the good DLA results of near 99% accuracy for both abdominal and spinal CT are promising. The DLA commonly showed approximately 94% of per-pixel row sensitivity in the three groups, which means that the DLA could not detect approximately 6% of the pixel row that contained metallic implants. The pixel rows missed by the DLA were mainly due to false classification of sacroiliac extension of the spine implants as hip implants. In addition, the DLA was unable to detect spine implants that lacked a classic rod-screw instrumentation system. On the abdominal CT AP topogram, the per-patient specificity of the DLA (99.5%) was significantly lower than that of the radiologists despite a 99.9% per-pixel row specificity. The DLA also incorrectly interpreted high-density materials such as residual barium or vertebroplasty. Based on these results, further training on the prototype DLA is necessary.

The diagnostic performance of the DLA for spine implants was excellent and not significantly different from either reader for both AP and lateral spinal CT topograms. Therefore, the direction of the topogram does not seem to affect the DLA performance for spine implants. However, the DLA showed lower specificity (94.6%) and extremely low PPV (37.2%) for hip implants. The main cause for this low specificity and PPV seems to be the false positive detection of sacroiliac extension of the spine implants as hip implants. Moreover, the DLA could not detect hip implants on lateral topograms at all. The primary reason for this was that lateral topograms were not included in the training phase for hip implants. Lateral spinal CT topograms generally cover only the cranial part of the hip implants unlike AP topograms. However, axial images for the lumbar spine are usually not extended to the hip joint level, and therefore, failing to detect hip implants from spinal CT lateral tomograms is not problematic from a practical standpoint. It would be better for the DLA to use AP topograms for CT examinations covering the hip joint.

We used CT topograms to develop the DLA in this study. As topograms are low-dose and low-resolution images, image quality is inferior to radiographs in detecting subtle objects. Some papers about automatic segmentation or detection of a certain part of the human anatomy have been published; however, there have been no studies on applying DLA to the detection of metallic implants in topograms [16,17]. As topograms are always obtained to localize and determine the range of CT scans, extracting useful information from topograms may help to perform CT examinations with more suitable parameters. Based on our results, the prototype DLA can streamline the radiographers’ tasks of choosing implant-specific iMAR preset and setting the range of the scan where the specific iMAR is selectively applied, even if the patient has more than one implant. Subsequent research is necessary to validate the usefulness in clinical practice of implementing the DLA to scan and reconstruct images in order to streamline the process of applying iMAR.

There are several limitations in this study. First, the DLA was developed and validated for the images acquired from multiple CT machines manufactured by only one vendor. Therefore, we cannot guarantee that the algorithm would show similar results for CT images to other vendors’ machines. Second, we evaluated the IoU for the pixel row, not for the exact segmentation of the metallic implant, even though the DLA provided the segmentation results. However, this approach is more practical because CT images that require iMAR are determined based on the range of metallic implants along the z-axis. Third, we were unable to implement the DLA as part of the iMAR workflow within the CT console to evaluate how beneficial the DLA would be in real clinical practice. Intensive validation should be performed before clinical implementation.

In conclusion, we developed a prototype DLA to detect metallic implants of the spine and hip on abdominal and spinal CT topograms. In the external validation, the DLA showed good performance for both spine and hip implants. However, it showed limited performance for hip implants on lateral topograms.

## Figures and Tables

**Figure 1 diagnostics-14-00668-f001:**
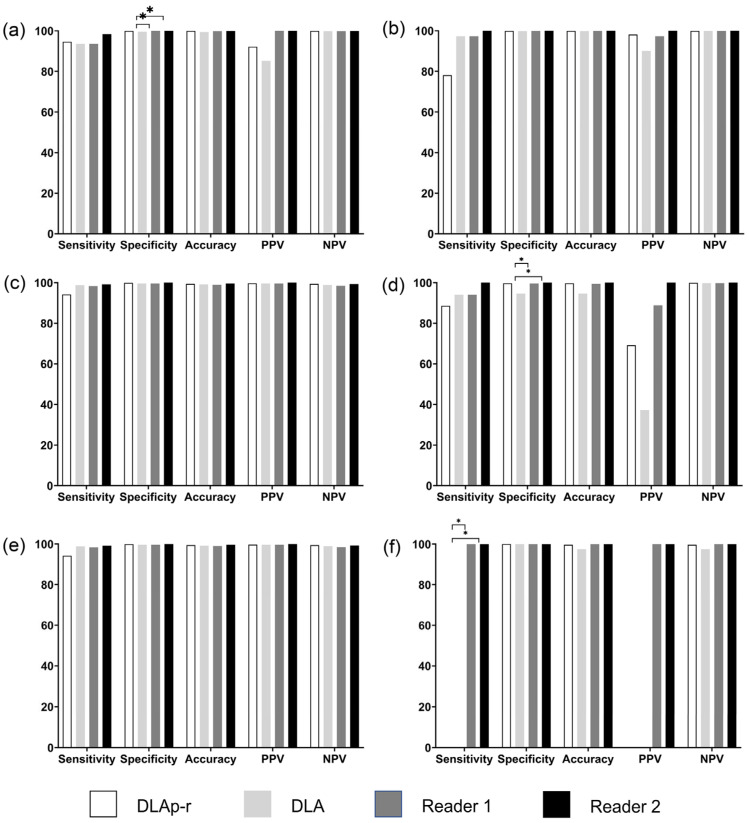
Results of the external validation. External validation results for spine implants and hip implants in abdominal_AP (**a**,**b**), spinal_AP (**c**,**d**) and spinal_Lat (**e**,**f**) topograms. * represents *p*-value < 0.05 in McNemar test. DLAp-r, deep-learning-based algorithm per pixel row; DLA, deep-learning-based algorithm; PPV, positive predictive value; NPV, negative predictive value.

**Figure 2 diagnostics-14-00668-f002:**
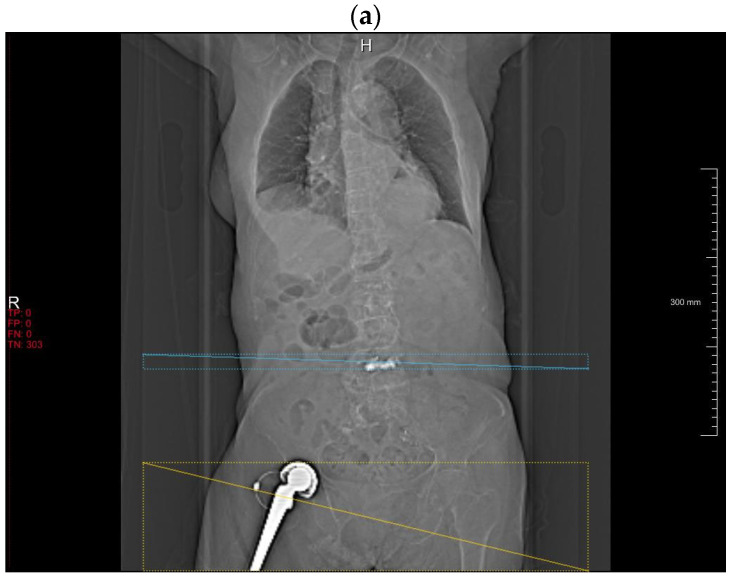
Falsely interpreted cases by the DLA. Bone cement in the spine is misinterpreted as spine implant (**a**). Sacroiliac extension of the spinal instrumentation is misinterpreted as hip implant (**b**). Non-conventional spinal implant (arrow) without typical rod-screw system is not detected (**c**).

**Figure 3 diagnostics-14-00668-f003:**
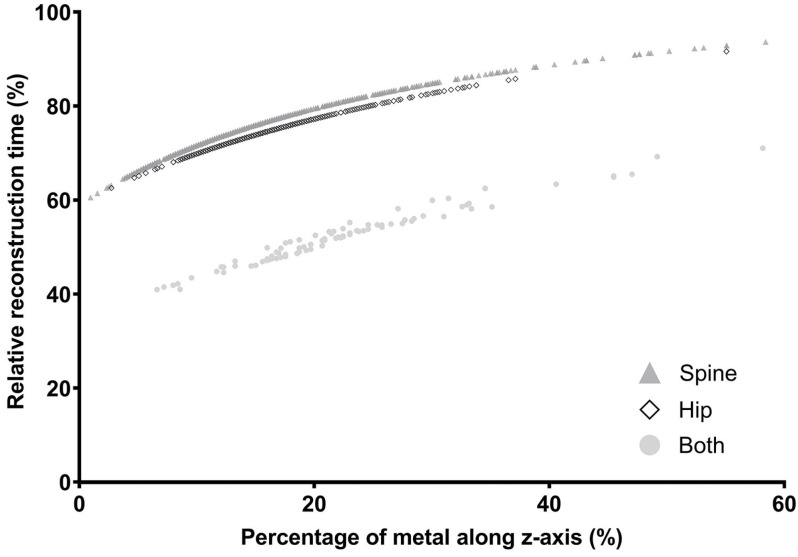
Relative time-saving by implementing the DLA to iMAR reconstruction.

**Table 1 diagnostics-14-00668-t001:** Diagnostic performance of deep-learning-based algorithm in internal validation.

	Spine Implants		Hip Implants	
	Per Pixel Row	Per Patient	Per Pixel Row	Per Patient
Sensitivity	96.9%	99.9%	97.9%	100%
Specificity	99.4%	95.6%	99.9%	98.4%
Accuracy	99.1%	98.6%	99.7%	98.9%
PPV	95.9%	98.2%	98.2%	96.4%
NPV	99.6%	99.8%	99.8%	100%

Note. PPV, positive predictive value; NPV, negative predictive value.

**Table 2 diagnostics-14-00668-t002:** Characteristics of external validation data.

	Abdominal CT_AP (*n* = 2178)		Spinal CT_AP (*n* = 515)		Spinal CT_Lat (*n* = 515)	
Projection	Anteroposterior		Anteroposterior		Lateral	
Age ^a^	61 [51–70]		70 [63–75]		70 [63–75]	
Sex (M:F)	1134:1044		193:322		193:322	
^b^ Scanner (1:2:3:4:5:6:7)	576:253:37:562:367:73:310		129:87:0:77:69:11:142		129:87:0:77:69:11:142	
Implants	Spine	Hip	Spine	Hip	Spine	Hip
No. of positive cases (both)	62 (4)	37 (4)	244 (10)	17 (10)	238 (6)	13 (6)
^a^ Craniocaudal length (number of pixels in z-axis)	55 [34–72.5]	84 [74–117]	84 [62–123]	412 [355–444]	87 [62–124]	62 [55–76]

Note. ^a^ Median [interquartile range] ^b^ 1: SOMATOM Force, 2: SOMATOM Force, 3: SOMATOM Force, 4: SOMATOM Definition AS+, 5: SOMATOM Definition AS+, 6: SOMATOM Definition, 7: SOMATOM Definition Edge.

**Table 3 diagnostics-14-00668-t003:** Intersection over union in true positive cases.

	Abdominal CT_AP		Spinal CT_AP		Spinal CT_Lat	
	Spine Implants (*n* = 56)	Hip Implants (*n* = 30)	Spine Implants (*n* = 239)	Hip Implants (*n* = 15)	Spine Implants (*n* = 239)	Hip Implants (*n* = 15)
DLA—Reader 1	* 0.939 [0.927–0.950]	0.977 [0.958–0.996]	* 0.936 [0.924–0.948]	* 0.912 [0.860–0.965]	* 0.933 [0.920–0.946]	NA
DLA—Reader 2	0.955 [0.945–0.966]	0.966 [0.948–0.984]	* 0.947 [0.938–0.956]	* 0.914 [0.856–0.973]	* 0.931 [0.918–0.943]	NA
Reader 1—Reader 2	0.960 [0.951–0.968]	0.978 [0.968–0.988]	0.969 [0.961–0.978]	0.973 [0.962–0.984]	0.981 [0.978–0.983]	0.983 [0.974–0.991]

Note. Data in brackets are 95% confidence intervals. * represents *p*-value < 0.05 in paired *t*-test between DLA-reader and interreader values. AP, anteroposterior; DLA, deep-learning-based algorithm; NA, not applicable.

## Data Availability

The data presented in this study are available on request from the corresponding author. The data are not publicly available due to the personal information protection act.

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
