# Peer review of "Development and Validation of a Deep-Learning-Based Algorithm for Detecting and Classifying Metallic Implants in Abdominal and Spinal CT Topograms"

_diagnostics, 2024, doi:10.3390/diagnostics14070668_

Round 1

Reviewer 1 Report

Comments and Suggestions for Authors

Dear authors,

I have read and appreciated your project. Below are my considerations.

The introduction is well-written and concise. The methods are clearly articulated. The statistical analysis is sufficient. The results are well-presented and summarized with tables. The figures are of adequate quality and provide valid support to the text. The discussion is well-exposed, giving the appropriate weight to the strengths and limitations of the study. The conclusions are well-written. The bibliography is appropriate. I would like to request clarification and a minor adjustment: At line 73 and subsequently in the text, what do you mean by "hospital A" and "hospital B"? Regarding Figure 1, could you enlarge the legends for better visibility?

Thank you.

Reviewer 2 Report

Comments and Suggestions for Authors

It seems that a lot of work has been done, but judging by the presented paper, there are a lot of white spots in it. And that’s spoil the article. So, the list of remark is:

The architecture of the artificial network is not clear. Was the input data 3-dimensional, or was it reduced to 2-dimensional or even 1-dimensional? Was the size of the input data normalized? Was only one convolution layer used! Which convolution kernels were used? Was the input data transformed during network training? How the threshold value for classification was found?

It is difficult to judge the results and discussion without a clear presentation of the materials and methods.

Round 2

Reviewer 2 Report

Comments and Suggestions for Authors

The authors have revised the article as required. I only advise to improve the quality of Figures 1 and 3. The figure labels are small. It is difficult to distinguish markers in figure 3.

Author Response

Thank you for your comments. Here is our response.

Comment) The authors have revised the article as required. I only advise to improve the quality of Figures 1 and 3. The figure labels are small. It is difficult to distinguish markers in figure 3.

Response: We have enlarged the labels on the figures for clearer delineation.